# Towards Effective Data-Free Knowledge Distillation via Diverse Diffusion Augmentation

Muquan Li
muquanli2023@std.uestc.edu.cn
University of Electronic Science and
Technology of China
Chengdu, Sichuan, China

Dongyang Zhang*
University of Electronic Science and
Technology of China
Chengdu, Sichuan, China
dyzhang@uestc.edu.cn

Tao He
University of Electronic Science and
Technology of China
Chengdu, Sichuan, China
tao.he01@hotmail.com

Xiurui Xie
University of Electronic Science and
Technology of China
Chengdu, Sichuan, China
xiexiurui@uestc.edu.cn

Yuan-Fang Li
Monash University
Melbourne, Victoria, Australia
yuanfang.li@monash.edu

Ke Qin
University of Electronic Science and
Technology of China
Chengdu, Sichuan, China
qinke@uestc.edu.cn

## Abstract

Data-free knowledge distillation (DFKD) has emerged as a pivotal technique in the domain of model compression, substantially reducing the dependency on the original training data. Nonetheless, conventional DFKD methods that employ synthesized training data are prone to the limitations of inadequate diversity and discrepancies in distribution between the synthesized and original datasets. To address these challenges, this paper introduces an innovative approach to DFKD through diverse diffusion augmentation (DDA). Specifically, we revise the paradigm of common data synthesis in DFKD to a composite process through leveraging diffusion models subsequent to data synthesis for self-supervised augmentation, which generates a spectrum of data samples with similar distributions while retaining controlled variations. Furthermore, to mitigate excessive deviation in the embedding space, we introduce an image filtering technique grounded in cosine similarity to maintain fidelity during the knowledge distillation process. Comprehensive experiments conducted on CIFAR-10, CIFAR-100, and Tiny-ImageNet datasets showcase the superior performance of our method across various teacher-student network configurations, outperforming the contemporary state-of-the-art DFKD methods. Code will be available at: https://github.com/SLGSP/DDA.

## CCS Concepts

• **Computing methodologies → Computer vision tasks**.

## Keywords

Data-Free Knowledge Distillation; Diffusion Model; Self-Supervised Data Augmentation

*Corresponding author

**ACM Reference Format:**
Muquan Li, Dongyang Zhang, Tao He, Xiurui Xie, Yuan-Fang Li, and Ke Qin. 2024. Towards Effective Data-Free Knowledge Distillation via Diverse Diffusion Augmentation. In *Proceedings of the 32nd ACM International Conference on Multimedia (MM '24), October 28-November 1, 2024, Melbourne, VIC, Australia.* ACM, New York, NY, USA, 10 pages. https://doi.org/10.1145/3664647.3680711

## 1 Introduction

Model compression is an essential task that seeks to reduce the size and complexity of deep models while maintaining their performance and functionality [42]. This becomes particularly significant in an era where large models are extensively used across a variety of platforms while computational resources are frequently limited. In the current wide variety of model compression technologies, knowledge distillation [17, 19, 52] stands out as a pivotal approach, which facilitates the transfer of knowledge from a complex and resource-intensive model, commonly known as the teacher model, to a more lightweight model referred to as the student model.

Despite knowledge distillation has demonstrated substantial success across multiple domains, the conventional approach typically necessitates access to the original data used to train the teacher model [7, 17]. Acquiring such training data can be exceptionally challenging due to its high cost and privacy concerns [3], which highlights the urgent demand for alternative approaches to training data acquisition. In response, recent literature has endeavored to address the issue of data scarcity by utilizing synthetic data in place of the original training data, a method referred to as data-free knowledge distillation (DFKD) [2, 4, 5].

Specifically, DFKD represents a fundamental paradigm within knowledge distillation, comprising two interrelated steps: synthesizing data that emulates the distribution of the original training data and distilling knowledge from teacher model to student model, as shown in Fig. 1. The first aspect, data synthesis, is typically divided into two distinct strategies: noise optimization [2, 29, 45] and generative reconstruction [4, 5, 9, 46]. Noise optimization involves using optimization algorithms to modify the noise of the input data, whereas generative reconstruction harnesses a generator network to establish a mapping from low-dimensional noise to the intricate data manifold. Compared to the former, generative reconstruction

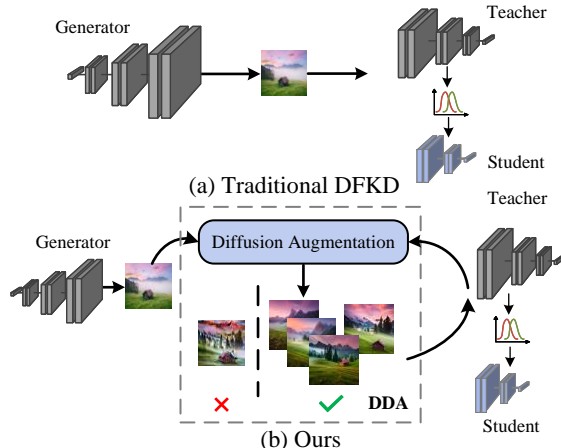

(a) Traditional DFKD

(b) Ours

**Figure 1: Comparison of the traditional DFKD with our method in terms of the overall framework.**

based methods alternately conduct data synthesis in each round of adversarial iteration, avoiding substantial time consumption and guarantees the production of high-quality data [23].

Nevertheless, current synthetic methods based on generative reconstruction are still inadequate in fulfilling several critical needs. To enable the student model to exhibit robust generalization across a multitude of tasks, the synthetic data must possess a richness and diversity that goes beyond mere replication of the original training data [9, 32]. Techniques such as image rotation for data augmentation in CSD [27] and channel-wise feature exchange (CFE) in SpaceshipNet [47] are employed to generate varied images. However, neither approaches can fully achieve the balance between the diversity and fidelity of the synthetic data inherent in adversarial network, where the number of synthesis must be constrained to maintain reliable domain prior [23], resulting in merely limited image generation while preserving the original image features. In addition, another concern is the necessity for the synthetic data to closely match the distribution of the original training data. Methods such as CutMix [48], Mixup [50], and SpaceshipNet [47] may inadvertently amplify unwanted noise in the synthesised image, consequently leading to potential distributional shifts in the final image set, as shown in Fig. 2. Therefore, adopting an updated approach to achieve alignment between synthetic and authentic training data is crucial for accurately capturing underlying patterns and complexities.

Recent researches have made it possible to generate richer visual representations with remarkable realism through the usage of diffusion models [1, 6, 43]. Specifically, these models excel at incorporating data with consistent semantics but varying information [43], thereby meeting the data augmentation needs in DFKD. However, the diffusion model may occasionally incur bias in a small number of cases during augmentation, causing spurious augmentation to certain extend. To mitigate this concern, continued investigation is essential to refine the augmentation process, effectively minimizing the occurrence of low-quality augmented images.

In this paper, we propose a novel DFKD method termed Diverse Diffusion Augmentation (DDA), which leverages diffusion models

to enrich the diversity of the generated data and employs cosine similarity-based filtering technique to ensure the fidelity of the augmentation process. Acknowledging the limitations of traditional generative reconstruction methods in low-quality data synthesis, we tackle the issue of noise amplification in synthetic images by deliberately reducing the impact of generative adversarial networks through model inversion [9, 45]. Instead of relying solely on single data synthesis, such process is expanded by incorporating a diffusion augmentation step, as shown in Fig. 1. In this setting, diffusion models are allowed to adaptively augment the images based on the semantic information understanding of student model within the images, thereby achieving a self-supervised data augmentation that enhances data diversity and constrain distribution bias simultaneously. To ensure augmentation fidelity, we further introduce cosine similarity to filter spurious augmentation. Referring to Fig. 3, through the integration of diffusion augmentation and cosine similarity-based filtering technique, our proposed method overcomes the inherent limitations of generative reconstruction, striking a balance between data diversity and fidelity, both pre- and post-augmentation. To comprehensively evaluate the generalizability and robustness of our proposed method, we conducted experiments on CIFAR-10, CIFAR-100 [6], and Tiny-ImageNet [22]. Extensive experiments provide concrete evidence that our method significantly outperforms state-of-the-art DFKD methods.

Our key contributions can be summarized as follows:

- We propose a novel DFKD method called DDA, which innovatively extends the conventional data synthesis in DFKD with data augmentation to further enhance data diversity, thus establishing a new paradigm for DFKD.
- We are the first to introduce diffusion models as a means of data augmentation in DFKD, which enriches the semantics and mitigates the distributional bias in synthetic data.
- To ensure the fidelity of the augmented images, we propose to use the cosine similarity method to filter out spurious augmentations.
- Experimental results confirm the effectiveness of our proposed DDA, showcasing superior performance when compared to the contemporary state-of-the-art DFKD methods.

## 2 Related Work

### 2.1 Data-Free Knowledge Distillation

The objective of DFKD is to transfer knowledge from a pre-trained teacher model to a student model without direct access to the original training data. Early approaches, exemplified by Lopes et al. [25], initially explore the utilizing of metadata from teacher model to reconstruct training data for knowledge distillation. Subsequent researches has largely moved away from metadata reliance and instead emphasizes alternative methodologies for knowledge transfer. One category of methods involves updating randomly initialized noise images by using optimisation algorithms to synthesize data. Nayak et al. [29] models the output of the teacher network as a Dirichlet distribution, while Yin et al. [45] regulates the distribution of synthesised images based on batch normalization statistics. Besides, CMI [9] asserts that data diversity enhances distillation performance and enables comparative learning to enhance instance

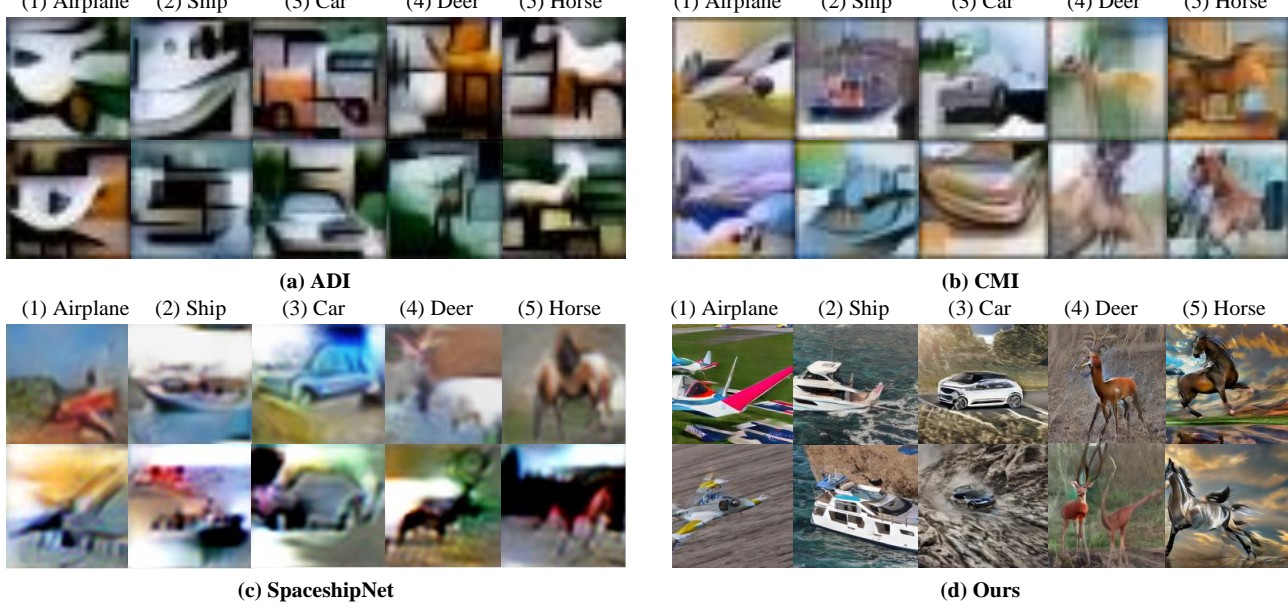

**Figure 2: The visualization of the synthesized data employed in the knowledge distillation training process for the pre-trained wrn-40-2 model on CIFAR-10. Three representative DFKD methods, ADI [45], CMI [9] and SpaceshipNet [47] are chosen to compared with our method. Obviously, our DDA is capable of achieving stronger instance distinguishability.**

discrimination. An alternative category of methods employs generative networks to synthesise training data [4, 5, 8, 9, 26, 28, 46]. These methods can be classified into non-conditional generative network-based methods [4, 5, 9, 26, 28] and conditional generative network-based methods [26, 46], depending on whether or not they combine conditional vectors when sampling random noise. Notably, Spaceshipnet [47] utilizes features from previous synthetic images to carry out channel-wise feature exchange (CFE) and employs multi-scale spatial activation region consistency (mSARC) as the constraint of similar network regions. Additionally, CDFKD-MFS [13] explores distilling knowledge from multiple teacher networks without access to original data, which uses a student network with additional parameters and multi-level feature-sharing to learn from multiple teachers. However, conventional DFKD methods are incapable of essentially achieving the balance of data diversity and distributional consistency inherent in generative reconstruction methods [23], causing less effective DFKD. In this paper, we propose a novel DDA method, which shifts the focus of synthetic data from adversarial networks to further data augmentation, overcoming inherent limitation in DFKD.

## 2.2 Data Augmentation

Data augmentation plays a crucial role in improving the robustness and generalisation capabilities of machine learning models across various domains. Traditional data augmentation techniques, as outlined in [39], typically involve fundamental transformations such as random flipping, cropping, and colour shifting, aimed at generating diverse versions of original images. Recent advancements in data augmentation have introduced mixup-based techniques[15, 20, 24, 30, 48, 50], which enhance diversity by blending patches of two

input images through convex combinations and adaptive sample mixing policies. Furthermore, generative models have emerged as a significant area of research in data augmentation, particularly in domains such as medical image augmentation [11, 36], domain adaptation [18], and bias mitigation [38]. These methods leverage trained or pre-trained Generative Adversarial Networks (GANs) [33] to generate images that adhere to desired distributions, which also facilitate dense visual alignment supervision and pixel-level annotation generation from limited labels. Notably, diffusion models [30, 31, 35, 51, 53] have demonstrated promise in generating training data in zero or few-shot settings, as well as producing challenging training examples. As research advances [34], it is anticipated that further exploration of diffusion models will enhance the effectiveness of data augmentation techniques. In this work, we re-visit the DFKD paradigm from another perspective, where the ability of data augmentation is utilized to enhance data diversity.

## 2.3 Diffusion Models

Diffusion models represent a significant advancement in generative models, offering remarkable capabilities in authentic image generation. While earlier methods like Variational Autoencoders (VAEs) [21] and GANs [33] laid the groundwork for realistic image synthesis, recent breakthroughs in this domain have predominantly been attributed to diffusion models. Demonstrated by [30, 31, 35, 51, 53], diffusion models have exhibited superior sample quality compared to traditional GAN-based approaches. Moreover, the evolution of diffusion models has paved the way for advancements in high-resolution image synthesis [34] and text-to-image generation [53], facilitated by innovations such as classifier-free guidance [41]. Diffusion models trained on large-scale datasets like LAION-5B [37]

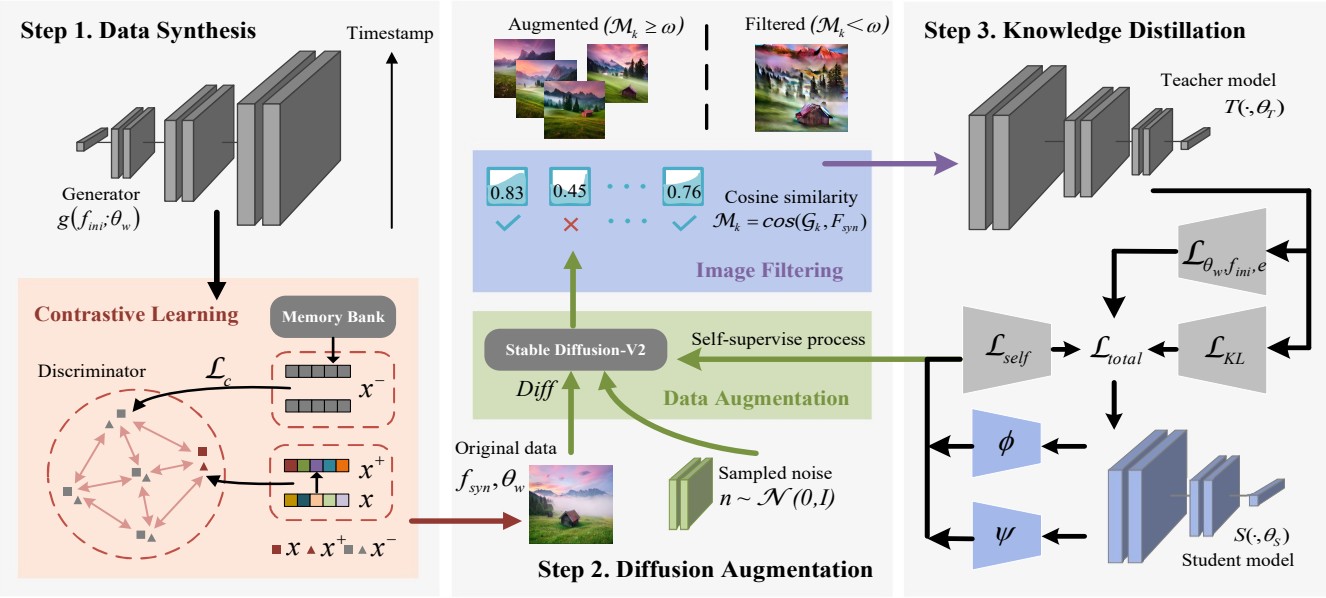

Figure 3: The illustrative framework of the proposed diverse diffusion augmentation (DDA) DFKD method. The three steps we present in the overall DFKD are arranged from left to right.

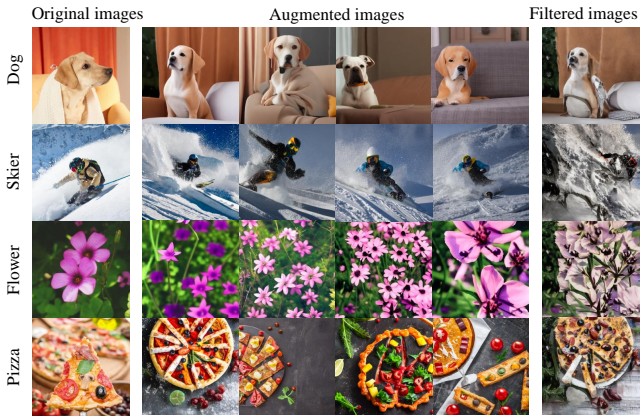

Figure 4: The visualization of diffusion augmentation and image filtering process. The diverse augmented and filtered low-quality images of several original images are shown.

have broadened their applicability across diverse domains, including point cloud generation and few-shot conditional image generation. Feng et al. [10] has emphasized the potential of diffusion models to enhance test-time prompt-tuning performance by directly incorporating semantically meaningful. Motivated by it, we utilize a powerful stable diffusion model to achieve effective data augmentation that maintain image semantics, as shown in Fig. 2.

## 3 Proposed Method

### 3.1 Preliminary

To achieve knowledge distillation, we initially give the definition of the teacher model $T(\cdot, \theta_T)$, the student model $S(\cdot, \theta_S)$, and the

original training dataset $D$ consisting of images $x \in \mathbb{R}^{H \times W \times C}$, where $H$, $W$ and $C$ refer to the height, width and channel number. For dataset $D$, $x_i$ and $x_j$ represent the image and the corresponding label, respectively. The distinctive aspect of DFKD compared to traditional knowledge distillation lies in enabling the student model $S$ to emulate the output of the teacher model $T$ for classification capability without direct access to $D$ [23]. To execute knowledge distillation without access to the original training dataset, model inversion is advisable to synthesize a $D'$ dataset with a distribution akin to that of $D$, which serves as the training data [9, 25].

Specifically, we first establish the model inversion to recover training data from a pre-trained model [9]. The proposed model inversion framework encompasses two fundamental components: the class prior $\mathcal{L}_{cls}$ and the Batch normalization (BN) regularization $\mathcal{L}_{bn}$. The class prior $\mathcal{L}_{cls}$ [4] is an one-hot assumption introduced in class-conditional generation, ensuring that the network predictions exhibit the same distribution as the original training data. It minimises the cross entropy (CE) loss of predefined labels and predictions from teacher model, as expressed below:

$$\mathcal{L}_{cls} = CE(x_j, T(\cdot, \theta_T)) \tag{1}$$

The BN regularization [45] process utilizes statistical information stored in the batch normalization layer of the teacher network as prior knowledge regarding the data. The regularization technique utilizes the running mean $\mu_l$ and running variance $\sigma_l^2$ of the $l$-th BN layer, which encapsulate the feature statistics of the original training data. The BN regularization is computed based on the disparity between the feature statistics of the synthetic data and those of the original training data. Mathematically, the BN regularization can be represented as the distance between the feature statistics

and the batch normalized statistics:

$$\mathcal{L}_{bn} = \sum_l \left( \|\mu_l(x) - \mu_l\|_2 + \|\sigma_l^2(x) - \sigma_l^2\|_2 \right) \tag{2}$$

where $\mu_l(x)$ and $\sigma_l^2(x)$ denote the mean and variance of the feature maps at the $l$-th BN layer, respectively.

Utilizing these two techniques, we introduce a unified inversion framework for data-free knowledge distillation:

$$\mathcal{L}_{in} = \alpha \cdot \mathcal{L}_{cls} + \beta \cdot \mathcal{L}_{bn} \tag{3}$$

where $\alpha$ and $\beta$ are parameters to balance $\mathcal{L}_{cls}$ and $\mathcal{L}_{bn}$, respectively. However, the aforementioned model inversion framework lacks consideration on data diversity and distributional consistency, which can result in the redundant generation of duplicate or irrelevant samples. To remedy this problem, we propose to improve the existing framework by incorporating contrastive learning and expand the data synthesis into a composite technique based on the diffusion model, as shown in Fig. 3.

## 3.2 Data Synthesis

Data diversity refers to the variability and distinctiveness within a training dataset [9]. A large amount of existing researches [9, 47] indicate that increased data diversity results in more robust instance discrimination and more effective knowledge distillation, even when the amount of data remains constant. Additionally, contrastive learning [44] represents a self-supervised technique that enables a neural network to learn the distinction between different instances, thereby serving as a appropriate metric for assessing data diversity.

*3.2.1 Contrastive Learning for Data Diversity.* A discriminator $disc$ is integrated into the contrastive learning framework, which is a multi-layer perception that takes the representation from the penultimate layer and the global pooling of intermediate features as input. A memory bank is introduced to store both historical and newly synthesized samples. For each image, a positive view $x^+$ is constructed using random augmentation, while other images in memory bank are considered as negative views $x_p^-$. The correct pairing of images from these positive and negative views necessitates a formulation to define the contrastive loss $\mathcal{L}_c$:

$$\mathcal{L}_c = -\mathbb{E}_{x_i \in \mathbb{R}} \left[ log \frac{exp\left(cos\left(x_i, x_i^+, disc\right)/tp\right)}{\sum_p exp\left(cos\left(x_i, x_p^-, disc\right)/tp\right)} \right] \tag{4}$$

where $tp$ signifies the temperature parameter of knowledge distillation to soften the distribution and $cos$ denotes the cosine similarity between individual data point $x$ in the new embedding space, which is mathematically expressed as the cosine of the angle between two vectors. The discriminator serves to assess data diversity by distinguishing between different views, thus pulling positive views closer together and pushing negative views further apart.

*3.2.2 Integration of Model Inversion.* The initial data synthesis task can be optimized through contrastive learning [44], seamlessly integrated into the foundational framework of model inversion. A generator $g$ and a memory bank $\mathcal{B}$ are introduced into the enhanced model inversion framework and merely one batch of data are synthesized by the generator $g$ in each timestamp $T$. At the

beginning of timestamp $T$, the teacher model is reinitialized for model inversion by generator $g$ and its initial potential features $f_{ini}$ along with the shared weights $\theta_w$ are iteratively optimized. Hence, the model inversion framework is structured as follows:

$$\mathcal{L}_{\theta_w, f_{ini}} = \alpha' \cdot \mathcal{L}_{in}\left(g\left(f_{ini}; \theta_w\right)\right) + \beta' \cdot \mathcal{L}_c\left(g\left(f_{ini}; \theta_w\right) \cup \mathcal{B}\right) \tag{5}$$

where $\mathcal{L}_{in}$ and $\mathcal{L}_c$ refer to the regular inversion framework in Eqn. 3 and the data diversity contrastive loss in Eqn. 4. Parameters $\alpha'$ and $\beta'$ are updated parameters utilized to balance these two losses, respectively. However, it is imperative to note that this inversion technique is constrained in achieving the balance of data diversity and distributional consistency [23]. To address this issue, the self-supervised diverse diffusion augmentation strategy is employed to further augment the available data.

## 3.3 Diverse Diffusion Augmentation

This section introduces the diverse diffusion augmentation technique for self-supervised data augmentation. For the first time, diffusion models are incorporated into DFKD methods not only to generate diverse and informative augmented images but also to maintain semantic consistency across various distribution. The Stable Diffusion-V2 [1] is selected as the decoder $Diff$ with sampled noise $n \sim \mathcal{N}(0, I)$, which is capable of generating the images $\mathcal{G}$ from the potential features $f_{syn}$ converted by a image encoder [1]. After model inversion, the obtained potential feature $f_{syn}$ of a single synthetic image is optimized and then passed through the diffusion model to generate the augmented images. Consequently, the augmented images can be represented as:

$$\mathcal{G}_k = Diff(f_{syn}, \theta_w, n) \tag{6}$$

where $\mathcal{G}_k$ denotes the $k$-th augmented image. However, employing a single augmented data for knowledge distillation may prompt the student model to repetitively synthesize samples, which may not be conducive to optimal student performance.

Therefore, we self-supervise this augmentation process, involving adaptively data augmentation guided by the comprehension abilities of students model, evaluated through its acquired semantic information from images in each training iteration. Specifically, the student model, typically a CNN model, comprises a feature extractor $\Phi$ with feature dimension $d$ and a classifier $\psi$. The augmented image is classified by the classifier, and the resulting classification is employed to compute the classification cross-entropy loss $\mathcal{L}_{self}$ based on the categories of the pre-augmented image:

$$\mathcal{L}_{self} = CE(x_j, \psi(\Phi(\mathcal{G}_k))) \tag{7}$$

where $\psi(\Phi(\mathcal{G}_k))$ denotes the predicted output of student model. The cross-entropy loss of this self-supervised task effectively reflects the ability of student model to comprehend the semantic information of the image, which can also contribute to enhancing the efficiency of data augmentation.

Additionally, in comparison to other augmentation methods, our images generated using the diffusion model exhibit higher resolution and diversity. Some examples of these images are illustrated in Fig. 2. Nevertheless, images generated solely from the potential features of the images may exhibit a certain degree of anisotropy, causing bite-sized portion of images to deviate excessively from the original images and diminishing the learning efficacy of the student

**Table 1: Experimental results of DFKD on CIFAR-10, CIFAR-100 and Tiny-ImageNet. Method $T$. and $S$. refer to the scratch training of teacher and student model on the labeled data.**

| Dataset | Teacher | Student | Test accuracy (%) | | | | | | | | |
|---|---|---|---|---|---|---|---|---|---|---|---|
| | | | T. | S. | DAFL | ZSKT | ADI | DFQ | CMI | SpaceshipNet | DDA |
| | resnet-34 | resnet-18 | 95.70 | 95.20 | 92.22 | 93.32 | 93.26 | 94.61 | 94.84 | 95.39 | **95.64** |
| | vgg-11 | resnet-18 | 92.25 | 95.20 | 81.10 | 89.46 | 90.36 | 90.84 | 91.13 | 92.27 | **93.02** |
| CIFAR-10 | wrn-40-2 | wrn-16-1 | 94.87 | 91.12 | 65.71 | 83.74 | 83.04 | 86.14 | 90.01 | 90.38 | **90.92** |
| | wrn-40-2 | wrn-40-1 | 94.87 | 93.94 | 81.33 | 86.07 | 86.85 | 91.69 | 92.78 | 93.56 | **93.63** |
| | wrn-40-2 | wrn-16-2 | 94.87 | 93.95 | 81.55 | 89.66 | 89.72 | 92.01 | 92.52 | 93.25 | **93.51** |
| | resnet-34 | resnet-18 | 78.05 | 77.10 | 74.47 | 67.74 | 61.32 | 77.01 | 77.04 | 77.41 | **77.56** |
| | vgg-11 | resnet-18 | 71.32 | 77.10 | 57.29 | 34.72 | 54.13 | 68.32 | 70.56 | 71.41 | **72.04** |
| CIFAR-100 | wrn-40-2 | wrn-16-1 | 75.83 | 65.31 | 22.50 | 30.15 | 53.77 | 54.77 | 57.91 | 58.06 | **58.96** |
| | wrn-40-2 | wrn-40-1 | 75.83 | 72.19 | 34.66 | 29.73 | 61.33 | 61.92 | 68.88 | 68.78 | **69.31** |
| | wrn-40-2 | wrn-16-2 | 75.83 | 73.56 | 40.00 | 28.44 | 61.34 | 59.01 | 68.75 | 69.95 | **70.27** |
| Tiny-ImageNet | resnet-34 | resnet-18 | 66.44 | 64.87 | N/A | N/A | N/A | 63.73 | 64.01 | 64.04 | **64.13** |

**Table 2: Ablation study on the threshold value of the cosine similarity filtering.**

| Threshold value | CIFAR-10 | CIFAR-100 |
|---|---|---|
| $\omega$=0.65 | 90.71 | 58.81 |
| $\omega$=0.7 | 90.86 | 58.85 |
| **$\omega$=0.75** | **90.92** | **58.96** |
| $\omega$=0.80 | 90.89 | 58.90 |
| $\omega$=0.85 | 90.85 | 58.88 |

model. To preserve the fidelity of the data augmentation, we reuse the cosine similarity measure discussed in Sec. 3.2.1 to filter out augmented images that show excessive deviation before and after augmentation. In specific, an additional mask $\mathcal{M}_k$ is introduced to determine whether to retain the $k$-th augmented image, based on its similarity exceeding a threshold parameter $\omega$:

$$\mathcal{M}_k = cos(\mathcal{G}_k, F_{syn}) > \omega \qquad (8)$$

where $F_{syn}$ represents the previous synthetic image by model inversion corresponding to the potential feature $f_{syn}$. This filtering process eliminates spurious augmentations generated by the diffusion model, effectively striking a balance between the diversity and fidelity of the augmented data. The outcomes of the filtering are visualized in Fig. 4, showcasing the improved quality of the augmented dataset.

## 3.4 Knowledge Distillation

In the stage of knowledge distillation, a final prediction loss is synthesised to serve as a constraint for the student model to emulate the output of the teacher model. The Kullback-Leibler (KL) divergence is commonly employed to minimize the logarithm of the outputs of both the teacher and student models:

$$\mathcal{L}_{KL} = KL(T(\cdot, \theta_T)/\tau, S(\cdot, \theta_S)/\tau) \qquad (9)$$

Additionally, feature maps are frequently employed to evaluate the loss of feature mapping between the two models. Furthermore, our self-supervised diffusion augmentation method integrates the

**Algorithm 1:** DFKD via Diverse Diffusion Augmentation.
___
**Input:** Pre-trained teacher model: $T$ and student model $S$.
**Output:** Optimized student model $S$.
___
Initialize prior knowledge $\theta_w$ and $f_{ini}$;
**while** not converged **do**
    Sample batch of noise $n$;
    **foreach** $epoch\ in\ Diff$ **do**
        Generate synthetic data $x_i$ with $\mathcal{L}_{\theta_w, f_{ini}}$;
        Implement data augmentation $\mathcal{G}_k \leftarrow Diff$;
        Evaluate the ability of obtaining semantic
          information of $S$ by $\mathcal{L}_{self} \leftarrow \Phi, \psi$;
        Filter invalid augmented images using $\mathcal{M}_k$;
    Store augmented data $\mathcal{G}_k$;

    **foreach** $epoch\ in\ knowledge\ distillation$ **do**
        Sample augmented data from $\mathcal{G}_k$;
        Evaluate the ability of classification $x_j \leftarrow \psi$;
        Optimize student network $S$ using $\mathcal{L}_{total}$;
    **return** Optimized student model $S$.

image recognition ability of the student model into image augmentation, implementing a form of adversarial learning. Consequently, the total knowledge distillation loss can be mathematically expressed as:

$$\mathcal{L}_{total} = \eta_{KL}\mathcal{L}_{KL} + \eta_{\theta_w, f}\mathcal{L}_{\theta_w, f} + \eta_{self}\mathcal{L}_{self} \qquad (10)$$

where $\eta_{KL}$, $\eta_{\theta_w, f, e}$ and $\eta_{self}$ are the parameters to balance three loss terms. The complete process and algorithm of DDA are outlined in Fig. 3 and Alg. 1, respectively. In summary, DDA enables the provision of more knowledge from the teacher model, thus enhancing distillation efficiency.

## 4 Experiments

In this section, we present a comprehensive set of experiments for DFKD, aiming to validate the efficacy of our proposed DDA method. We start by outlining the experimental setup, including the specific tools and configurations employed throughout the experimentation process. Next, we benchmark the performance of

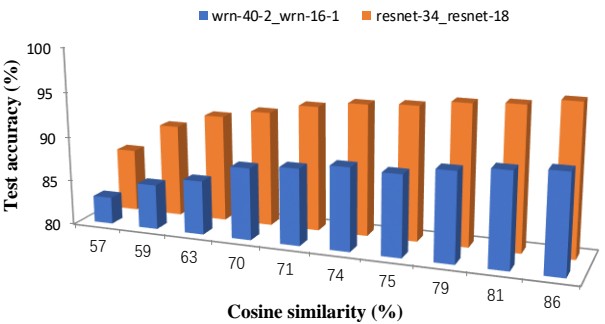

**Figure 5: The influence of the cosine similarity on two teacher-student networks, wrn-40-2 to wrn-16-1 and resnet-34 to resnet-18. The positive correlation tendencies demonstrate the positive effect of cosine similarity on the results.**

DDA against state-of-the-art DFKD methods to assess its potential as a promising paradigm in the field. Additionally, we conduct ablation studies to analyze the significance and effectiveness of the different components within our method.

## 4.1 Settings

*4.1.1 Models and Datasets.* We implement knowledge distillation on several different models, including resnet [14], vgg [40], and wide resnet (wrn) [49]. Three popular classification datasets, namely CIFAR-10, CIFAR-100 [6], and Tiny-ImageNet [22] are utilized as benchmarks to evaluate several existing DFKD methods. CIFAR-10 and CIFAR-100 each consist of 60,000 images with 32×32 resolution, of which 50,000 are used for training and the remaining 10,000 for testing. Alternatively, Tiny-ImageNet has 64×64 resolution images, encompasssing 100,000 training images and 10,000 test images. These three types of datasets also contain 10, 100 and 200 classes, respectively.

*4.1.2 Diffusion models.* The data augmentation step occurs after generating data through model inversion. We employed Stable Diffusion-V2 [1] to augment the data obtained from the synthetic images, expanding three new images for each synthetic image. The guidance scale of the Stable Diffusion-V2 and the number of diffusion steps are set to 0.5 and 50 respectively.

*4.1.3 Implementation details.* All teacher models mentioned above are trained on labelled datasets, whereas the student models are trained on data generated by inversion of the teacher model. During model inversion, we update the generator using the Adam optimizer with a learning rate of 1e-3, which in turn synthesises 200 images per step, with 500 repetition steps. Subsequently, the student model is trained using a SGD optimizer with a learning rate of 0.1 and a momentum of 0.9, with cosine annealing decay of 1e-4.

## 4.2 Comparison with State-of-the-arts Methods

Table 1 elaborately presents the results of DFKD for state-of-the-art (SOTA) methods in recent years, including DAFL [4], ZSKT [28], ADI [45], DFQ [5], CMI [9] and SpaceshipNet [47], as well as our DDA. According to the classification of DFKD methods, DAFK, ZSKT, DFQ, CMI, SpaceshipNet, and DDA belong to the

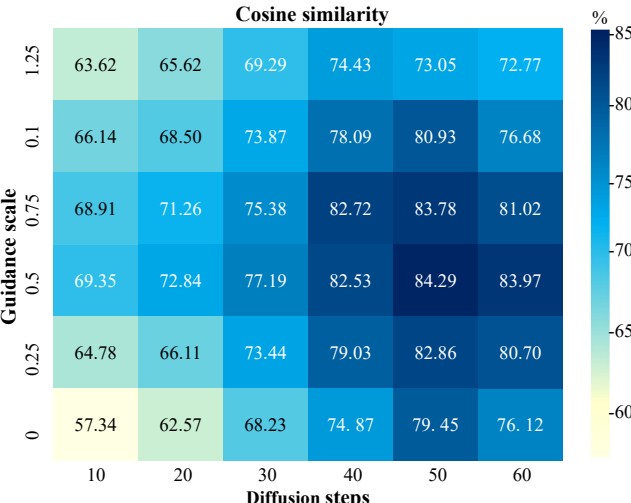

**Figure 6: Variation of the cosine similarity versus the guidance scale and number of diffusion steps.**

generative reconstruction type of DFKD, while ADI belongs to the noise optimization type. The main difference between our method and the baselines methods is the usage of a diffusion model to extend the data synthesis. For a fair comparison, we utilize the same teacher-student networks as visual backbones for all experiments. The results in Tab. 1 evidence that DFKD method guided by DDA quantitatively outperforms the current SOTA methods on all three commonly used datasets by a large margin.

In addition, Fig. 2 further visualizes the data utilized in knowledge distillation training process of four methods: ADI [45], CMI [9], SpaceshipNet [47] and DDA. It is evident that our augmented images exhibit considerably clearer and more restorative characteristics with better diversity. The visualization also distinct character evidently that DeepInversion produces images with similar colour and texture, while the CMI method improves highly image diversity but neglects sufficient clarity and resolution. Additionally, SpaceshipNet uses CFE network to enhance image content but still falls short of enabling recognition with the naked eye. In contrast, the images produced by DDA are readily recognizable even without magnification, facilitating an effective capture of the features and objects present in each image.

## 4.3 Ablation Study

*4.3.1 The influence of cosine similarity.* Before investigating the effects of other parameters on the distillation results, it is imperative to rigorously establish the positive correlation of cosine similarity with the final outcomes. An experiment was conducted on the CIFAR-10 dataset, employing wrn-40-2 to wrn-16-1 and resnet-34 to resnet-18 as the two teacher-student networks. Fig. 5 illustrates that the growth of cosine similarity improves test accuracy, indicating that a augmented image dataset with higher cosine similarity yields superior results.

*4.3.2 The effect of guidance scale and diffusion steps.* This section investigates the influence of two critical parameters in the diffusion model, diffusion steps and guidance scale [1], which are

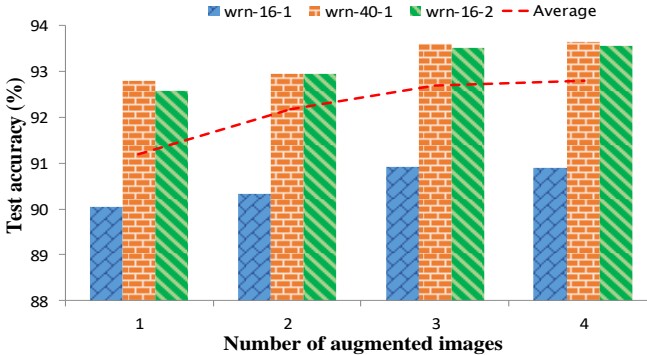

**Figure 7: Ablation study on the augmented dataset scale.**

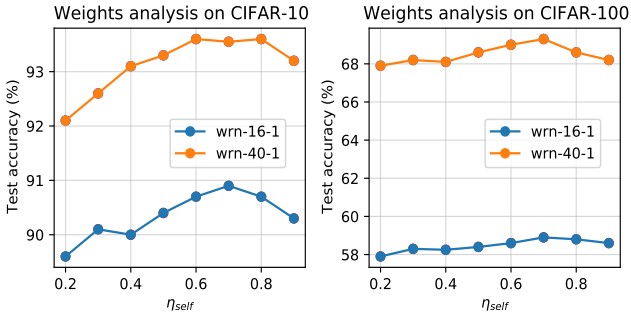

**Figure 8: Ablation study on the weights of DDA loss.**

related to the cosine similarity of the augmented dataset. In a stable diffusion model, each diffusion step predicts noise from a Gaussian distribution, and the prediction of model relies more on the source condition as the guidance scale increases. Fig. 6 illustrates that the diffusion model, when applied for augmenting our synthetic data approach, produces highest quality images with 50 diffusion steps and guidance scale of 0.5.

*4.3.3 **The effect of threshold parameter $\omega$.*** As stated in Sec 3.3, $\omega$ is utilized to filter augmented images below a specific cosine similarity threshold. We evaluated the results of using different values of $\omega$ on the CIFAR-10 and CIFAR-100 datasets using wrn-40-2 to wrn-16-1 as the teacher-student model network. Tab. 2 reveals that threshold parameter of 0.75 enhances knowledge distillation performance. A reasonable explanation is that a small threshold may result in low-quality knowledge being transferred, while a large threshold may decrease the diversity of data and constrain the generalisation of the student model. Such value of 0.75 is exactly moderate for the threshold, striking a balance that results in the best performance.

*4.3.4 **The effect of the scale of the augmented dataset.*** We further explore the impact of augmented dataset scale on the results, building upon the experiments in Sec 4.3.3. As depicted in Fig. 7, it is apparent that the testing accuracy exhibits remarkable growth with the increase in augmented data scale, while the growth in accuracy becomes marginal during the transition from 3 × to 4 × augmented images, and the incorporation of more augmented images undoubtedly diminishes the extent of model compression. To strike a balance between accuracy and model size, our work

**Table 3: Fréchet Inception Distance (FID) of augmented data. We estimate FID metric in three shallow layers, namely the 1-st Pool, the 2-nd Pool, and the deep final Pool.**

| Feature position | 1-st Pool | 2-nd Pool | Final Pool |
|---|---|---|---|
| WGAN-GP [12] | N/A | N/A | 29.3 |
| ADI [45] | 2.021 | 17.28 | 84.69 |
| CMI [9] | 0.140 | 1.776 | 62.63 |
| Ours | **0.127** | **1.532** | **56.33** |

**Table 4: Ablation study about the effect of diffusion model and cosine similarity.**

| Method | DM. | CS. | CIFAR-10 | CIFAR-100 |
|---|---|---|---|---|
| (a) | X | X | 89.33 | 57.65 |
| (b) | ✓ | X | 90.69 | 58.74 |
| (c) | X | ✓ | 89.54 | 57.85 |
| (d) | ✓ | ✓ | **90.92** | **58.96** |

augments each synthesized image into three images. Moreover, we estimate the average of Fréchet Inception Distance (FID) [16] score for tripled number of augmented images, where a lower score indicates higher quality of images. Referring to the Tab. 3, our augmented images achieve the lowest score across all three layers, demonstrating their superior diversity from low-level features to high-level semantics.

*4.3.5 **The effect of diffusion model and cosine similarity.*** The effects of two core modules in DDA: the diffusion model and the cosine similarity are investigated, where we disable each module both in conjunction and individually to evaluate its influence. As shown in Tab. 4, the test accuracy decreases by an average of 1.245% when the diffusion model is deactivated, and by an average of 0.225% when the cosine similarity is disabled. This indicates that the diffusion model has a more significant impact on the results, with its deactivation leading to a more substantial degradation in accuracy. Moreover, Fig. 8 shows the effect of parameter $\eta_{self}$ in Eqn. 10 where the term $\mathcal{L}_{self}$ is proposed by ourselves. It can be observed that the model's accuracy nearly consistently improves with the increase of the $\eta_{self}$, which demonstrates the effectiveness of our diffusion model and cosine similarity.

## 5  Conclusion

In this paper, we pioneer the application of diffusion models in the domain of DFKD, achieving enhanced data diversity with reduced distributional bias. Additionally, to mitigate redundancy in data augmentation, we introduce a image filtering technique based on cosine similarity, which eliminates augmented images exhibiting significant deviations pre- and post-augmentation, resulting in improved performance. Through extensive experimentation on three widely-used datasets and various teacher-student model pairs, we have achieved state-of-the-art results, highlighting the high effectiveness of our DDA method. As diffusion models and large-scale models continue to evolve rapidly, investigating alternative diffusion models for data augmentation in knowledge distillation offers a promising direction for future research.

# Acknowledgments

This work was partially supported by grants from the Postdoctoral Fellowship Program of CPSF ( No. GZB20240115), National Natural Science Foundation of China (No. 62306064) and the Research Funding of Science and Technology on Information System Engineering Laboratory (No. 6142101230202).

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
