# OpenReview forum: "Towards Effective Data-Free Knowledge Distillation via Diverse Diffusion Augmentation"
_acmmm.org/ACMMM/2024/Conference — MM2024 Poster_

### Official Review · Reviewer_HGQZ · 2024-05-10

**Rating:** 3
**Confidence:** 3

**Summary:**

This work focuses on DFKD (Data-Free Knowledge Distillation) and tries to solve the limitations of conventional DFKD methods: synthesized training data are prone to the limitations of inadequate diversity and discrepancies in distribution between the synthesized and original datasets. This work tries to use diffusion models to solve the above challenges through diverse diffusion augmentation (DDA). To be specific, they revise the paradigm of common data synthesis in DFKD to a composite process through leveraging diffusion models subsequent to data synthesis for self-supervised augmentation and introduce an image filtering technique grounded in cosine similarity to maintain fidelity during the knowledge distillation process.

**Strengths:**

1.	The proposed method is the first to empower DFKD with diffusion models.
2.	The proposed method achieves the best performance across 3 datasets compared with the baselines.

**Limitations:**

1.	The proposed method uses cosine similarity to maintain the fidelity of diffusion-generated images during the process of DFKD and uses an ablation study in Table 2 to obtain the optimal threshold. Would it be better if the authors could show us some examples (image pairs) with different cosine similarities? In that case, it could be more vivid for us to have a general idea about what kind of generated images are suitable for the distillation.
2.	Generally, as the authors stated in this work, the limitations of DFKD lie in the inadequate diversity and discrepancies in distribution between the synthesized and original datasets, which could be attributed to the inaccessibility of the training data. I totally agree with this statement. However, the proposed method directly adopts Stable-Diffusion-V2, which is trained with mountains of data collected from the Internet (maybe ImageNET included). As a result, if the diffusion model was trained with the datasets the authors use in their experiments, I am afraid that it violates the fundamental assumption of DFKD problems – the inaccessibility of training data, making the process non-DFKD. Consequently, I could hardly agree that it is fair to compare the proposed method with DFKD methods.
3.	The results of Figure 5 and Table 2 seem confusing. Figure 5 offers a 3D view of the influence of the cosine similarity on two teacher-student networks, whereas I would prefer a 2D chart with the accuracy annotated. Besides, from it, we can conclude that the larger the cosine similarity is, the better the test acc is. This conclusion seems to contradict the results in Table 2, which suggests that a too-high threshold harms the final accuracy. Could the author further clarify these two results?

**Suitability:**

2

---

### Official Review · Reviewer_aChR · 2024-05-28

**Rating:** 3
**Confidence:** 3

**Summary:**

This paper introduces a diffusion model for data synthesis in DFKD to increase the diversity and fidelity of synthetic data.

DFKD is divided into two steps: data synthesis to emulate the original data (training data for a teacher model) and knowledge distillation from the teacher model to the student model. In particular, this paper focuses on the former.

Experiments show that the proposed DDA is superior to the state-of-the-art DFKD method.

**Strengths:**

1. This is the first study to incorporate a diffusion model into DFKD data synthesis.

2. Experiments have shown the effectiveness of the proposed method and an analysis of the influence of hyperparameters. In particular, the effectiveness of the new ideas ($\mathcal{L}_{\text{self}}$ and the filtering by cosine similarity) are also analysed.

**Limitations:**

1. Most of the framework appears to rely on CMI [9]. For example, the procedures in Sec. 3.1 and 3.2 would be already described in the literature [9], although with some variations.

2. Some lack of explanation of the proposed method. If a part of the method follows CMI [9], it should be clearly stated to enhance reproducibility. The following questions also remain:
* What kind of network is $\textit{disc}$ in Sec. 3.2.1 (line 477)?
* How is $f_{\text{syn}}$ in Sec. 3.3 (line 525) is obtained? It is stated that “the potential feature $f_{\text{syn}}$ converted by a image encoder [43]”, but there is no mention of "image encoder" or "potential features" in the reference [43].
* What is $\mathcal{L}_{\theta{w}, f}$ in Eq. (10) (line 631)?

3. About novelty. The most important point is how much the idea of incorporating a diffusion model into data synthesis in DFKD contributes to this field. Although not in the field of DFKD, several studies have already attempted to synthesize training data using the diffusion model [1*][2*].

[1*] He, Ruifei, et al. "IS SYNTHETIC DATA FROM GENERATIVE MODELS READY FOR IMAGE RECOGNITION?." Proceedings of the International Conference on Learning Representations. 2022.

[2*] Shipard, Jordan, et al. "Diversity is definitely needed: Improving model-agnostic zero-shot classification via stable diffusion." Proceedings of the IEEE/CVF Conference on Computer Vision and Pattern Recognition. 2023.

**Suitability:**

2

---

### Official Review · Reviewer_GonV · 2024-05-30

**Rating:** 3
**Confidence:** 3

**Summary:**

This paper introduces a Data-Free Knowledge Distillation (DFKD) method named DDA. DDA aims to address the diversity-fidelity dilemma in data synthesis in DFKD, with proposed diffusion augmentation and cosine similarity filtering. Experiments show the effect of DDA in generating high-fidelity samples in DFKD, which results in a better distillation performance and beats previous SOTA.

**Strengths:**

The proposed method is a novel application of diffusion models in DFKD. The idea is attractive, and intuitively, the idea makes sense and has potential.

Proven by the experiments, the performance is attractive.

The quality of the manuscript is above average.

**Limitations:**

Flaws in presentation.

1) While the majority of the paper is easy to follow, some important parts of the proposed method are confusing. For example, in section 3.2.1, how the positive pairs and negative pairs are generated is not clear. Figure 3 indicates $x^-$ comes from a memory buffer, while there are not any descriptions in section 3.2.1. Also, in line 479, the "random transformations" are not clear. It is recommended to be specific.

Flaws in equations.

1) The equations use the same symbols for different variables/values, and it can be misleading. For example, $x_j$ is used as labels in line 419 and Eqn. (1), while it is used as negative images in Eqn. (4). Another example is that $\tau$ is used as the temperature parameter of contrastive learning in Eqn. (4), while it is used as the temperature of KL divergence in Eqn. (9).

2) It might be a misunderstanding, but in line 547, $x_j$ should be the label, not $\uppsi(\cdot)$.


More information about experiments is suggested.

1) Since the author claims DDA is better than other methods in generating high-resolution images (in line 777). It is suggested to include experimental results on high-resolution datasets, for example, ImageNet or its subset. Such experiments would be valuable to the paper and could better ground the claim.

2) It is suggested that more information about the experiment be included to help reviewers assess the quality of the experiments. For example, the number of runs of the experiment conducted for a given setting and the error bars are important information. Also, it is suggested to include the training time for each method.

Summary:
The idea is great, and the quality of the paper is above average. However, I do have some questions/concerns. I'm glad to update the score if the concerns are properly addressed.

**Suitability:**

3

---

### Meta-Review · Area_Chair_vhYs · 2024-06-26

**Recommendation:** Accept (Poster)
**Confidence:** 3

**Metareview:**

This paper received two borderline accept and one borderline reject, making it a borderline paper. After the rebuttal process, many of the concerns have been addressed, and the authors have promised further modifications. Therefore, I lean towards accepting this paper. Despite some remaining concerns, the proposed method shows strong potential and achieves excellent performance in data-free knowledge distillation, contributing to the field.